# Deep autoencoder-based behavioral pattern recognition outperforms standard statistical methods in high-dimensional zebrafish studies

Adrian J. Green[1,2]*, Lisa Truong[3], Preethi Thunga[1], Connor Leong[3], Melody Hancock[1], Robyn L. Tanguay[3], David M. Reif[1,4]

1 Bioinformatics Research Center, Department of Biological Sciences, NC State University, Raleigh, North Carolina, United States of America, 2 Sciome LLC, Research Triangle Park, North Carolina, United States of America, 3 Department of Environmental and Molecular Toxicology, Oregon State University, Corvallis, Oregon, United States of America, 4 Predictive Toxicology Branch, Division of Translational Toxicology, National Institute of Environmental Health Sciences, Durham, North Carolina, United States of America

* ajgreen4@ncsu.edu

**Data Availability Statement:** The code and data required to replicate findings reported in the article are available at https://github.com/Tanguay-Lab/

## Abstract

Zebrafish have become an essential model organism in screening for developmental neurotoxic chemicals and their molecular targets. The success of zebrafish as a screening model is partially due to their physical characteristics including their relatively simple nervous system, rapid development, experimental tractability, and genetic diversity combined with technical advantages that allow for the generation of large amounts of high-dimensional behavioral data. These data are complex and require advanced machine learning and statistical techniques to comprehensively analyze and capture spatiotemporal responses. To accomplish this goal, we have trained semi-supervised deep autoencoders using behavior data from unexposed larval zebrafish to extract quintessential "normal" behavior. Following training, our network was evaluated using data from larvae shown to have significant changes in behavior (using a traditional statistical framework) following exposure to toxicants that include nanomaterials, aromatics, per- and polyfluoroalkyl substances (PFAS), and other environmental contaminants. Further, our model identified new chemicals (Perfluoro-n-octadecanoic acid, 8-Chloroperfluorooctylphosphonic acid, and Nonafluoropentanamide) as capable of inducing abnormal behavior at multiple chemical-concentrations pairs not captured using distance moved alone. Leveraging this deep learning model will allow for better characterization of the different exposure-induced behavioral phenotypes, facilitate improved genetic and neurobehavioral analysis in mechanistic determination studies and provide a robust framework for analyzing complex behaviors found in higher-order model systems.

Manuscripts/tree/main/Green_et_al_(2024)_Manuscript.

**Funding:** This research was supported by the National Institutes of Health (NIH) grant awards ES030287 (RLT, LT), ES030007 (AJG, DMR), ES025128 (DMR), ES033243 (DMR), and CA161608 (AJG, DMR). This research was supported [in part] by the Intramural Research Program of the NIH, ZIAES103385 (DMR). The funders played no role in the study design, data collection and analysis, decision to publish, or preparation of the manuscript.

**Competing interests:** The authors have declared that no competing interests exist.

## Author summary

We demonstrate that a deep autoencoder using raw behavioral tracking data from zebrafish toxicity screens outperforms conventional statistical methods, resulting in a comprehensive evaluation of behavioral data. Our models can accurately distinguish between normal and abnormal behavior with near-complete overlap with existing statistical approaches, with many chemicals detectable at lower concentrations than with conventional statistical tests; this is a crucial finding for the protection of public health as exposure can lead to a range of neurodevelopmental disorders, including cognitive and other behavioral deficits. Our deep learning models enable the identification of new substances capable of inducing aberrant behavior, and we generated new data to demonstrate the reproducibility of these results. Thus, neurodevelopmentally active chemicals identified by our deep autoencoder models may represent previously undetectable signals of subtle individual response differences. Our method elegantly accounts for the high degree of behavioral variability associated with the genetic diversity found in a highly outbred population, as is typical for zebrafish research, thereby making it applicable to multiple laboratories generating similar data. Utilizing the vast quantities of control data generated during high-throughput screening is one of the most innovative aspects of this study and to our knowledge is the first study to explicitly develop a deep autoencoder model for anomaly detection in large-scale toxicological behavior studies.

## Introduction

Significant progress continues to be made in our understanding of neurodevelopmental disorders such as autism spectrum disorder, attention deficit hyperactivity disorder (ADHD), developmental delay, learning disabilities, and other neurodevelopmental problems. As incidences continue to rise globally and affect 10–15% of all births, more work must be done to improve our understanding of these disorders [1–3]. Meta-analyses suggest strong and consistent epidemiological evidence that the developing nervous system is particularly vulnerable to low-level exposure to widespread environmental contaminants, as the anatomical and functional architecture of the human brain is mainly determined by developmental transcriptional processes during the prenatal period [3–7]. Therefore, identifying associations between developmental exposures and neurological effects is a core objective to improve public health by informing disease and disability prevention [1,8].

As the number of environmental contaminants grows to nearly one million, comprehensive data on the neurodevelopmental toxicity of these contaminants remain sparse or nonexistent [3,9–11]. In response, high-throughput screening (HTS) assays have been developed to expedite chemical toxicity testing using *in vitro* and *in vivo* systems [12–14]. However, *in vitro* cell and cell-free assays cannot fully capture systemic organismal responses in terms of anatomy, physiology, or behavior [15]. Zebrafish (*Danio rerio*) have emerged as an ideal model for studying low-level chemical exposure because of their high fecundity, rapid development, genetic tractability, and amenability to high-throughput data generation [12,16,17]. The zebrafish brain's structural organization, cellular morphology, and neurotransmitter systems are very similar to other vertebrates, including chickens, rats, and humans [18–21]. Furthermore, zebrafish have behavioral patterns highly similar to mammals, and genetic homologs for 70% of human genes and 82% of human disease genes, making them a powerful model organism for revealing the neuronal developmental pathways underlying behavior [22–24].

Zebrafish larvae show swimming patterns essential for their survival following swim bladder development at four to five days post-fertilization (dpf), including exploration, foraging, and escape response which can be assessed using various locomotor behavioral assays [25,26] while more advanced continuous swimming, schooling, and reproductive behavior is still developing. One of these assays, the larval photomotor response (LPR), utilizes a sudden transition from light to dark, eliciting a stereotyped large-angle O-bend, followed by several minutes of increased movement, which gradually reduces [27,28]. Exposure to toxicants has been shown to alter this stereotypical behavioral response [24,29]. Current HTS for behavioral neurotoxicity focuses heavily on analyzing locomotor behavior using distance moved and population-based statistical methods [24,30]. However, while the behavior repertoire of larval zebrafish is less sophisticated when compared to that of adult zebrafish and other higher-order vertebrates, they are capable of numerous distinct behaviors [24,31,32]. These behaviors, such as thigmotaxis, and light avoidance cannot always be captured when using distance moved as a sole indicator of neurobehavioral toxicity in analyses of this data. Moreover, as most laboratory zebrafish populations feature significant genetic heterogeneity, individual responses to exotic toxicants cannot be expected to be homogeneous for simplistic measures such as distance moved [33].

Improved accessibility to computing resources and application interfaces, together with recent advances in deep-learning makes it possible to analyze complex behavioral data in novel ways and predict neurodevelopmental toxicity [34–36]. The volume and diversity of data generated during HTS experiments, combined with the variety in toxicological response within populations, present an opportunity that is well-suited for machine learning (ML). In particular, analysis of zebrafish HTS data from five dpf larvae exposed to 1,060 unique chemicals reveals that only 8% of chemical-concentration pairs (a unique combination of chemical and concentration, e.g. 6.4 μM Nicotine) exhibit changes in distance moved [30], which is low given the known toxicity profiles of the chemical set. The traditional methods for analyzing zebrafish behavior data are primarily based on measurement of distance moved and instances of variations in the movement patterns, velocity changes and spatial preference is lost due to the sheer volume of data and complexity. Additionally, the traditional analysis methods is unable to identify meaningful patterns due to the noise and variability. This challenge provides an opportunity to apply methods developed for anomaly detection from areas such as financial fraud [37], medical application faults [38], security systems intrusion [39], system faults [40], and others [41,42]. Such ML techniques would allow for a more holistic evaluation of zebrafish behavior by learning complex features such as movement patterns, velocity changes and spatial preferences associated with "normal" behavior and flagging subtle deviations. These intricate nuances could be indicative of chemical toxicity and can often be missed by traditional assays relying solely on measuring distance moved as a metric. In anomaly detection, we learn the pattern of a normal process, and anything that does not follow this pattern is classified as an anomaly. This learning model is particularly applicable, as many HTS data sets have large amounts of control data to analyze [30]. One intriguing approach to achieving this is by applying an autoencoder [43–48]. An autoencoder is a neural network of two modules, an encoder and a decoder [47,49]. The encoder learns the underlying features of a process, and these features are typically in a reduced dimension. The decoder then uses this reduced dimension to recreate the original data from these underlying features.

In the present study, we trained deep autoencoder models to recognize the pattern of quintessential larval zebrafish behavior and identify abnormal behavior following developmental chemical exposure. The performance of our deep autoencoders was compared against a two sample Kolmogorov–Smirnov test (K-S test), a standard for behavioral assessment. In addition to model development, we assessed the features driving performance through feature

permutation and generated new confirmatory data to assess model reproducibility and confirm novel findings.

## Results

### Statistical classification of behavior

A two sample Kolmogorov–Smirnov test (K-S test) was used to compared treated vs control distance moved and angular velocity in light/dark cycling in zebrafish larvae at five dpf. We identified 40 chemical-concentration combinations from nine chemicals and 28 chemical-concentration combinations from nine chemicals capable of inducing a significantly different ($p < 0.05$) behavioral response using both distance moved and angular velocity, respectively (S2 Table). While 10 chemicals were identified using both methods, nine chemicals were similar, with distance moved finding a significant difference in multi-walled carbon nanotubes at 75 and 100 μM and angular velocity finding a difference in sodium 2-(N-ethylperfluorooctane-1-sulfonamido)ethyl phosphate at 0.25 μM. Considering that distance moved revealed more chemical-concentration combinations in this screening application, we used this metric to identify abnormal larvae to ensure a sufficient number for training the autoencoder models. Using the 30th and 70th percentiles, we defined 227 individual larvae as abnormal (Fig 1A). These 227 larvae formed the validation set used to test the performance of our models.

### Training performance

Autoencoder models were trained using only control data for each of the activity states (hypoactive, normal, and hyperactive) per phase of the second light cycle. This resulted in six trained models (S1 Fig the training loss plots for the models). Table 1 shows the results for the six deep autoencoder models trained using control data and validated using data from zebrafish defined as abnormal using the K-S test. All the models performed well with values ranging from 0.615–0.867 and 0.740–0.922 for the Kappa and AUROC, respectively. As expected, the models consistently produced high specificity (SP) levels as this value indicated how well the models classify control data. There was greater variability in the sensitivity (SE) with the dark phase models matching or outperforming the light phase models for each activity state. Further, we observed a noteworthy trend across all models producing high positive predictive value (PPV). Overall, these results show that deep autoencoders trained using control data is capable of distinguishing between normal and abnormal larval zebrafish behavior with a high degree of accuracy.

### Evaluation of unknowns

Using the six trained models, we evaluated the 2,719 treated zebrafish larvae (Fig 1). The autoencoders correctly classified 156 of the 227 larvae that fell below or above the 30th and 70th percentiles, respectively. In addition, our deep autoencoders identified 463 larvae as abnormal from the 2,492 larvae defined as normal using the K-S test (Fig 1B). The majority (422) of these 619 larvae were from one of 66 chemical-concentration combinations from 13 chemicals (Table 2). The deep autoencoders successfully identified nine of the ten statistically abnormal chemicals and identified these chemicals at or below the lowest concentration shown to be statistically significant. While the deep autoencoders did not identify Perfluorodecylphosphonic acid as capable of inducing abnormal behavior, but they did identify 3-Perfluoropentyl propanoic acid (5:3), Perfluoro-n-octadecanoic acid, 8-Chloroperfluorooctylphosphonic acid, and Nonafluoropentanamide, which were missed in the statistical testing framework. These results,

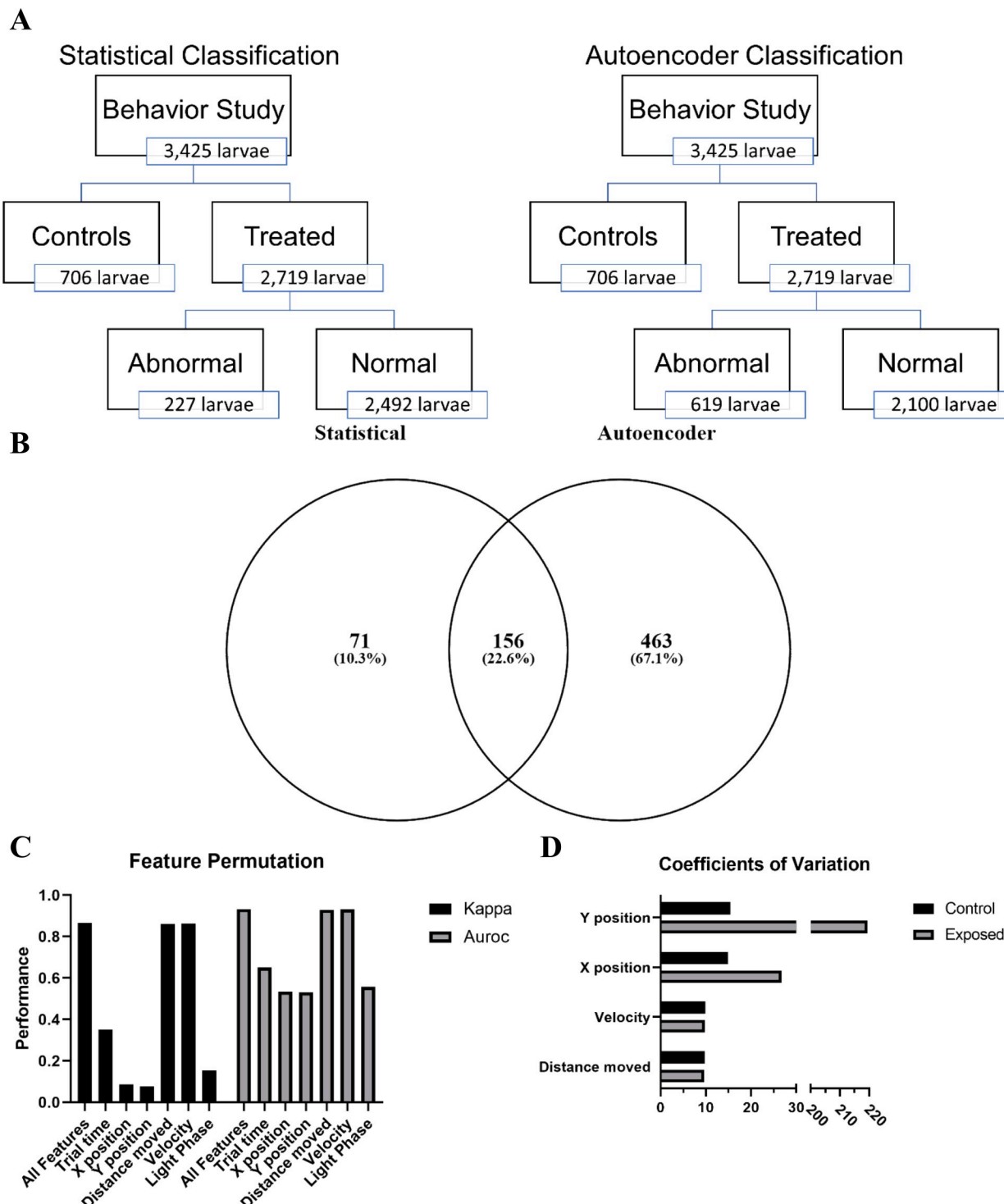

**Fig 1. Assessment autoencoder performance.** (A) Schematic representation of the differences in statistical and autoencoder based classification of behavioral response in larval zebrafish. (B) Venn diagram showing overlap between statistical and autoencoder classified abnormal zebrafish. (C) Evaluating the change in model performance when the values of a single feature are randomly shuffled. Kappa—Cohen's Kappa statistic, AUROC—area under the receiver operating characteristic. Figure depicts means ± SEM. (D) Coefficients of variation for each of the main numerical features.

**Table 1. Deep autoencoder model performance in behavioral classification.** Table showing performance of model trained using different activity states of the control data in both light and dark phases.

| Model | | Performance Metrics | | | | |
|---|---|---|---|---|---|---|
| **Baseline Control Activity Level** | **Light Phase** | **SE** | **SP** | **PPV** | **Kappa** | **AUROC** |
| Hypoactive | Light | 78.5 | 100 | 99.7 | 0.867 | 0.892 |
| | Dark | 78.3 | 98.0 | 88.4 | 0.800 | 0.882 |
| Normal | Light | 48.3 | 99.7 | 93.1 | 0.615 | 0.740 |
| | Dark | 73.3 | 94.8 | 77.6 | 0.695 | 0.840 |
| Hyperactive | Light | 79.2 | 97.5 | 85.5 | 0.790 | 0.883 |
| | Dark | 86.9 | 97.5 | 90.2 | 0.855 | 0.922 |

summarized in Fig 2, show that deep autoencoders can match the performance of the K-S test and are more sensitive at detecting abnormal behavior.

## Features driving improved autoencoder performance

To determine the features in the model that were most important in driving classification performance, we employed permutation feature importance. This technique is a model agnostic inspection technique used for any fitted estimator to determine the importance of each feature in the model. Larger the decrease in model performance (Kappa or AUROC) when a single feature value is randomly shuffled, the more important the feature. Our results, shown in Fig 1C, indicate that phase, trial time, x position, and y position are the largest drivers of model performance, while distance moved and velocity contribute very little. Coefficients of variation show greater variability in the x and y positional data between control and exposed groups compared to either velocity or distance moved (Fig 1D). This trend is consistent irrespective of the larval activity state (hypoactive, normal activity, or hyperactive) relative to their respective controls (Fig 3).

## Experimental confirmation of autoencoder findings

To provide an unbiased evaluation of the final model fits, we generated new data using 2-Methylphenanthrene, and Nonafluoropentanamide. The data collected confirmed that our

**Table 2. Autoencoders identified chemicals.** Table showing chemicals and concentrations flagged for displaying abnormal behavioral effects when evaluated using Autoencoder. Compounds that were picked up by Autoencoder, but not KS test are highlighted in red.

| CASRN | Chemical Name | Concentration (μM) |
|---|---|---|
| 71751-41-2 | Abamectin | 0.1, 0.2, 0.4, 0.6 |
| 308068-56-6 | Multi-Walled Carbon Nanotube | 10, 23.2, 50, 75, 100 |
| 2531-84-2 | 2-Methylphenanthrene | 1, 2.54, 6.45, 16.4, 35, 74.8, 100 |
| 832-69-9 | 1-Methylphenanthrene | 1, 2.54, 6.45, 16.4, 35, 74.8, 100 |
| 914637-49-3 | **3-Perfluoropentyl propanoic acid (5:3)** | 0.25 |
| 192-51-8 | Dibenzo[e-l]pyrene | 0.01, 0.025, 0.065, 0.164, 0.35, 0.75, 1, 2.54, 16.4, 35, 100 |
| 16517-11-6 | **Perfluoro-n-octadecanoic acid** | 0.25 |
| 355-46-4 | Perfluorohexanesulfonic acid | 0.015, 0.14, 0.41, 3.7, 11.1, 33.3, 66.5, 100 |
| 3834-42-2 | (Heptafluoropropyl)trimethylsilane | 0.015, 0.046, 0.41, 1.23, 11.1, 33.3 |
| | **8-Chloroperfluorooctylphosphonic acid** | 0.167 |
| 31253-34-6 | 2-Aminohexafluoropropan-2-ol | 0.015, 0.046, 0.41, 1.23, 3.7, 11.1, 33.3, 66.5, 100 |
| 13485-61-5 | **Nonafluoropentanamide** | 0.41, 3.7, 11.1 |
| 439-14-5 | Diazepam | 1, 3, 5, 8, 12 |

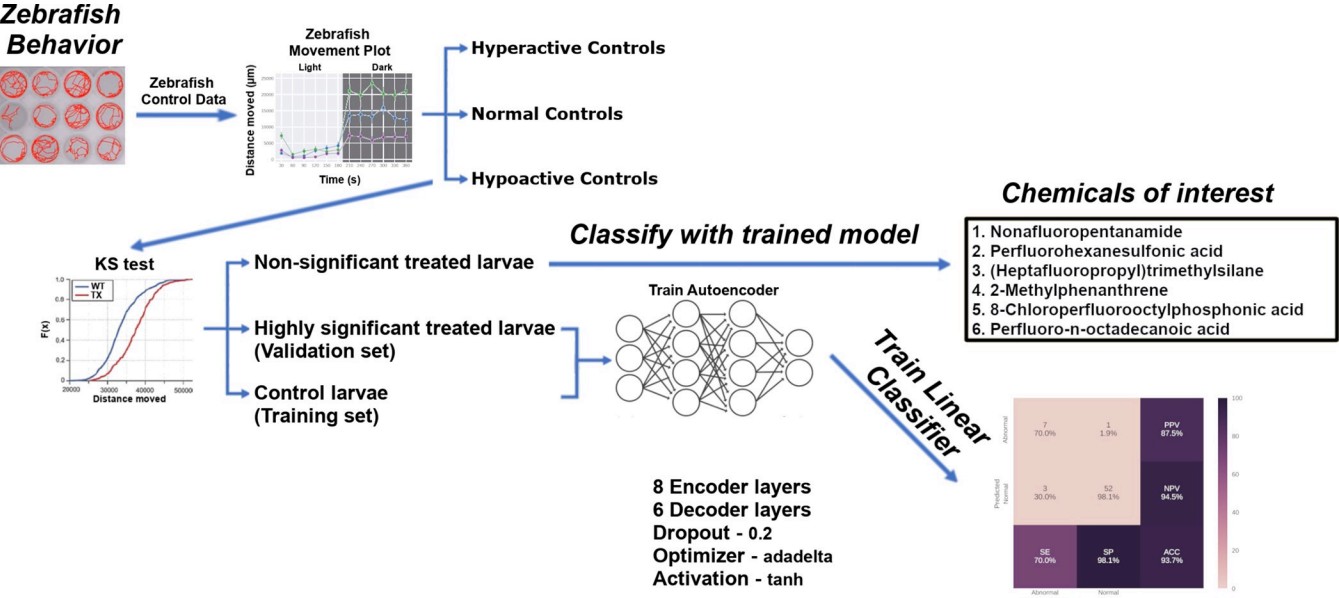

**Fig 2. Summary of behavioral analysis pipeline and results.** Utilizing our analysis pipeline produced six deep autoencoder models (three for the light phase and three for the dark phase) capable of classifying larval zebrafish behavior with high Kappa and AUROC values. The trained models were then used to classify the non-significant exposed larvae and identified Nonafluoropentanamide, Perfluorohexanesulfonic acid, (Heptafluoropropyl)trimethylsilane, 2-Methylphenanthrene, 8-Chloroperfluorooctylphosphonic acid, Perfluoro-n-octadecanoic acid, and others as capable of inducing abnormal behavior.

models accurately classified all controls as normal while detecting similar levels of abnormal behavior response across the concentration range (Fig 4) (p > 0.15). These results show that the trained model is capable of producing similar results across experimental replicates.

## Discussion

Statistical analysis identified 39 chemical-concentration combinations from ten chemicals capable of inducing a significantly different (p < 0.05) behavioral response. Utilizing the 227 abnormal individuals identified by the statistical test as our validation set, we trained six deep autoencoder models using control data for each of the activity states (hypoactive, normal, and hyperactive). All of the resulting models performed well with values ranging from 0.615–0.867 and 0.740–0.922 for the Kappa and AUROC, respectively. All models achieved SP values above 94.8% and PPV values above 77.6% while SE values for all dark phase models outperformed the light phase models for each activity state (Table 1). Assessment of permutation feature importance indicates that phase, trial time, x-position, and y-position are the largest drivers of model performance (Fig 1C). The calculated coefficients of variation shed some light on this surprising finding (Fig 1D). They show that variation in the x and y positional data is greater than observed for velocity or distance moved between control and exposed groups. These differences in variation likely make it easier for the models to distinguish between treated and exposed groups.

When we examined exposed larvae defined as normal using the K-S test (Fig 1), our deep autoencoders identified 66 chemical-concentration combinations from 12 chemicals (Table 2) with Perfluoro-n-octadecanoic acid, 8-Chloroperfluorooctylphosphonic acid, and Nonafluor-opentanamide only identified by our autoencoders. These results show that a deep autoencoder-based model can classify larval zebrafish behavior as normal or abnormal with very good efficacy and often identified abnormal behaviors at lower concentrations than current

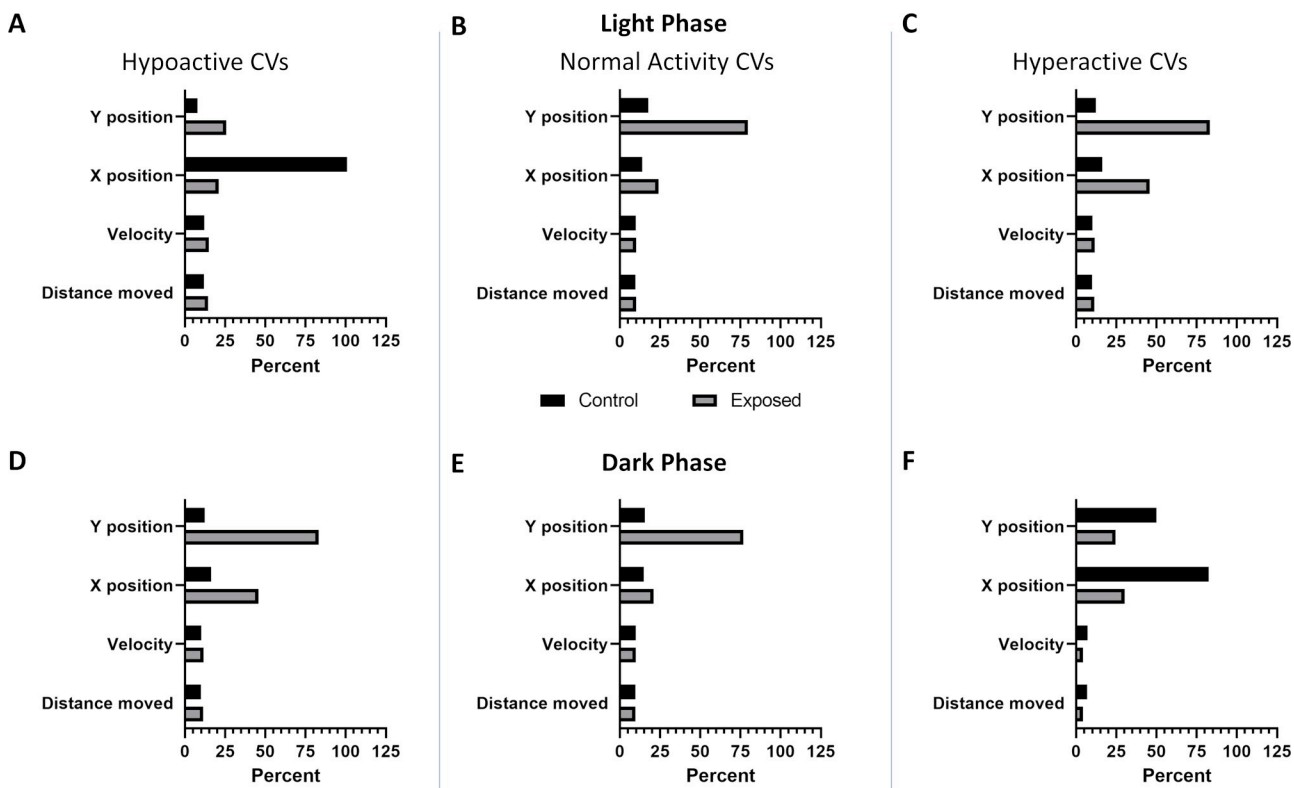

**Fig 3. Coefficients of variation per larval activity state.** Coefficients of variation (CVs) for each of the main numerical features (A–C) in the light (D–F) and in the dark. Columns show CVs of larval zebrafish significantly (p < 0.05) (A, D) hypoactive, (B, E) normal activity, or (C, F) hyperactive relative to their respective controls.

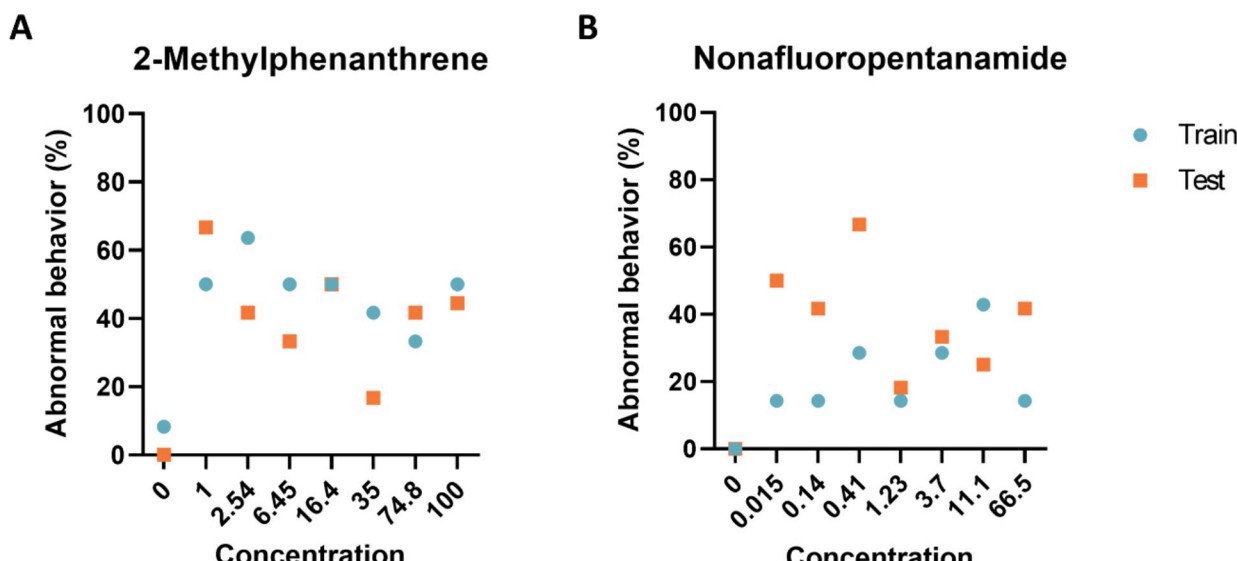

**Fig 4. Experimental model evaluation.** Comparison of the performance of deep autoencoder models between the training set and two chemicals identified by the models to elicit abnormal larval zebrafish behavior. Percent of larval zebrafish classified as abnormal based on their behavioral response to developmental exposure to (A) 2-Methylphenanthrene and (B) Nonafluoropentanamide.

statistical methods. Further, the models identified three novel chemicals, Perfluoro-n-octade-canoic acid, 8-Chloroperfluorooctylphosphonic acid, and Nonafluoropentanamide as capable of inducing abnormal behavior (Fig 3). While making a definitive claim will require further experimentation, it does appear that the autoencoder method is particularly sensitive at detecting changes due to PFAS exposure. PFAS are associated with increased glutamate levels in the hippocampus and catecholamine levels in the hypothalamus, decreased dopamine in the whole brain after PFAS exposure, and increased extracellular glutamate has been observed in the hippocampus epileptic rats [50,51]. Thus, it is reasonable to infer that these neurochemical changes are capable of altering autoencoder-detectable patterns without changing locomotor magnitude or direction.

Recognition and categorization of swimming patterns in larvae is a challenging task and a number of approaches have been used. These can range from subjective analysis based on experienced observations [31,52] or through the application of unsupervised ML [27,32,53–57]. These studies have focused on the analysis and categorization of behavioral patterns in wild-type strains [27,57], mutant strains [32,53], or larvae exposed to neuroactive chemicals [32] but do not classify behavior as normal or abnormal. In addition, these unsupervised approaches have utilized highspeed camera systems which are medium to low throughput and have limited potential in the screening of tens of thousands of chemicals for behavioral effects. As introduced above, classification of behavior is one of the primary goals of toxicological screening and tends to result in highly imbalanced datasets and lend themselves to anomaly detection methodologies. While these methods are common in manufacturing [41–43,58], information systems [38,40], security systems [39,45], and financial fraud [37] they have only very recently been applied to biological data [44,59,60]. To the best of our knowledge, this is the first study to explicitly develop a deep autoencoder model for anomaly detection in toxicological behavior studies.

Overall, our results show that a deep autoencoder utilizing raw behavioral tracking data from five dpf zebrafish larvae can accurately distinguish between normal and abnormal behavior. We show that these results are reproducible and allow for the identification of new compounds capable of eliciting abnormal behavior. Further, our models were able to identify abnormal behavior following chemical exposure at lower concentrations than with traditional statistical tests such as the two sample Kolmogorov–Smirnov test (K-S test). Our approach accounts for the high degree of behavioral variability associated with the genetic diversity found within a highly outbred population typical of zebrafish studies, thereby making it extensible to use across labs. Our deep autoencoders only needed seven hundred controls and a three-minute light and three-minute dark cycle to identify differences. The majority of zebrafish labs have historical or the ability to generate similar data that can be used to train their own deep autoencoder models. Looking to the future, neurodevelopmentally active chemicals identified using our deep autoencoder models may represent heretofore undetectable signals of subtle differences in individual responses, suggesting chemicals that should be investigated further as eliciting differential population responses (i.e. interindividual susceptibility differences).

These findings will facilitate the application of behavioral characterization methods discussed above, such as ZebraZoom [32], using highspeed cameras to identify the behavioral traits most perturbed by the chemical exposure and allow for more mechanistic discovery. One of the key innovations presented in this study is leveraging vast amounts of control data generated as part of any high-throughput screening (HTS)–setting the stage for predictive toxicological applications and safety assessments for the enormous backlog of as-yet untested chemicals.

## Materials and methods

This section describes the autoencoder models utilizing a semi-supervised ML algorithm and logistic regression (LR) to discriminate between normal and abnormal behavior in chemically exposed five dpf zebrafish. An overview of our approach is shown in Fig 2. Briefly, we created and trained six autoencoder models for each phase of the assay; namely, hyperactive, normal, and hypoactive depending on the control movement in the light or dark phases of the assay. Finally, treated plates were tested on one of these, depending on which category, its controls fell under. We used experimental data collected on a large and diverse compound set of 30 chemicals including an insecticide, nanomaterial, perfluorinated chemicals, and aromatic pollutants at a range of concentrations (133 chemical-concentration pairs) to assess the neurotoxic effects of these chemicals following developmental exposure (S1 Table).

### Ethics statement

This study was conducted in accordance with the guidelines and regulations set forth by the Institutional Animal Care and Use Committee (IACUC) at Oregon State University. The protocol was reviewed and approved by the IACUC under the approval number 2021–0227. All procedures involving animals were performed in compliance with the ethical standards of the institution and adhered to the principles of humane animal treatment.

### Zebrafish husbandry

Tropical 5D wild-type zebrafish were housed at Oregon State University's Sinnhuber Aquatic Research Laboratory (SARL, Corvallis, OR) in densities of 1000 fish per 100-gallon tank according to the Oregon State University Animal Use Care and Protocol: 2021–0227 [61]. Fish were maintained at 28˚C on a 14:10 h light/dark cycle in recirculating filtered water, supplemented with Instant Ocean salts. Adult, larval and juvenile fish were fed with size-appropriate GEMMA Micro food 2–3 times a day (Skretting). Spawning funnels were placed in the tanks the night prior, and the following morning, embryos were collected and staged [62,63]. Embryos were maintained in embryo medium (EM) in an incubator at 28˚C until further processing. EM consisted of 15 mM NaCl, 0.5 mM KCl, 1 mM $MgSO_4$, 0.15 mM $KH_2PO_4$, 0.05 mM $Na_2HPO_4$, and 0.7 mM $NaHCO_3$ [63].

### Developmental chemical exposure

The empirical data used to develop our model were gathered as described in Truong et al. and Noyes et al. [12,64,65]. The experimental design consisted of the 30 unique chemicals tested (S1 Table) with at least 7 replicates (an individual embryo in singular wells of a 96-well plate) at each concentration for each chemical. The concentrations evaluated were based on preliminary studies within the authors' lab to span lethal and sub-lethal concentration range were possible based on physical chemicals properties including solubility.

### Developmental toxicity assessments

**Mortality and morphology.**   At 24 hours post-fertilization (hpf), embryos were screened for mortality, developmental delay, and spontaneous movement [12]. At 120 hpf, mortality, craniofacial abnormalities (eye, snout and jaw), body axis abnormalities, edema (yolk sac and pericardial edema), upright body abnormalities (swim bladder, somite and circulation), touch response brain abnormalities (brain, otic vesicle and pectoral fin), pigment, notochord, and trunk abnormalities (trunk and caudal fin) [12,66,67]. The incidence of abnormality across all morphology endpoints were evaluated as binary outcomes. Any individuals identified with a

physical abnormality were excluded from any behavioral analysis as these abnormalities might confound the results.

**Photomotor responses.** The larval photomotor response (LPR) assay was conducted at 120 hpf when the 96-round well plates of larvae were placed into a Zebrabox (Viewpoint Life-Sciences) and larval movement was recorded. The recorded videos were then tracked with Ethovision XT v.11 analysis software for 24 min across 3 cycles of 3 min light: 3 min dark with an initial 6 minute dark acclimation period. The trial time(s), x-position, y-position, distance moved (μm), and velocity (mm/s) by each larva in the 2nd light/dark cycle were the features used for behavioral assessment (S2 Fig). The 2nd light/dark cycle was chosen as it exhibited less noise than the 1st cycle and was less influenced by any learning that might have occurred in the 3rd cycle. For all assessments, data were collected from embryos exposed to nominal concentrations of chemical and uploaded under a unique well-plate identifier into a custom LIMS (Zebrafish Acquisition and Analysis Program [ZAAP])–a MySQL database and analyzed using custom R scripts that were executed in the LIMS background [29].

## Data preprocessing and statistical analysis pipeline

**Preprocessing.** All data processing, statistical analysis and ML were implemented in Python using the open source libraries Tensorflow [68], Keras [69], Scikit-learn [70], Pandas [71], and Numpy [72] within a purpose build Singularity container environment [73]. The x-position and y-position data was standardized relative to the center of each well and forward filled if datapoints were missing. Outliers were normalized to the maximum likely distance a zebrafish larva could move in $1/25^{th}$ of a second. Considering that the average length of a 5 dpf larval zebrafish is 3.9 mm and can move about 2.5 times it's body length during a startle response (120 frames at 1000 frames/second) the threshold for distance moved in our system was set at 3.25 mm per frame [53,74]. This resulted in 5,445 of the 30,825,000 frames being normalized.

## Statistical analysis

A two sample Kolmogorov–Smirnov test (K-S test), a non-parametric two-sided test with no adjustments for normality or multiple comparisons, was used to compare each chemical-concentration combination with their respective same plate controls ($p < 0.05$). Interexperimental zebrafish larval response to light/dark cycling is highly variable (S2 Fig). Therefore, it was essential to group the unexposed controls based on the mean from individual 96-well plates compared to mean movement for unexposed controls across all plates. Controls from individual plates with statistically significant ($p < 0.01$) differences in movement compared to the average of all controls were grouped together as hyperactive, normal, or hypoactive. Following grouping the K-S test was used to compare Individuals in the $30^{th}$ and $70^{th}$ percentiles of each chemical-concentration combination were defined as abnormal.

## Autoencoder architecture

Deep autoencoders were developed using zebrafish control data to distinguish between normal and abnormal zebrafish behavior. The model was trained on a Dell R740 containing two Intel Xeon processors with 18 cores per processor, 512 GB RAM, and a Tesla-V100-PCIE (31.7 GB). The autoencoders consisted of an input and output layer of fixed-size based on the size of a single phase (25 frames per 180s) of the second light cycle (4500 frames by 5 features). The encoder network was composed of eight fully connected hidden layers using a normal kernel initialization, tanh activation, a dropout value of 0.2, L1 and L2 regularization values of $1e^{-05}$, and an adadelta optimizer. The size of each hidden layer was reduced by increasing multiples

of 15 and resulted in a compressed representation (bottleneck) size of 250. The decoder network was composed of six fully connected hidden layers using tanh activation, and a dropout value of 0.2. All hidden layers used an adadelta optimizer (learning_rate = 0.001, rho = 0.95, and epsilon = 1e-07) and mean squared error for the loss function [75–77]. For each model, we optimized the hyperparameters (i.e., the number of hidden layers, the number of nodes in the layers, loss functions, optimizers, regularization rates, and dropout rates) by grid search technique trained on all control data over 500 epochs using Cohens Kappa statistic as the objective metric. The final encoder models were trained over the course of 125000 epochs. The resulting compressed representation was used as input into a logistic regression layer trained using a 100 fold cross-validation with each fold consisting of 4000 epochs using a limited-memory BFGS solver. The code and dataset are available at GitHub [https://github.com/Tanguay-Lab/Manuscripts/tree/main/Green_et_al_(2024)_Manuscript].

## Network performance and evaluation

The data showed strong normal vs abnormal class imbalance (Fig 1). Classifiers may be biased towards the major class (normal) and therefore, show poor performance accuracy for the minor class (abnormal) [78]. Normal vs abnormal classification accuracy was evaluated using a confusion matrix, Cohen's Kappa statistic, and area under the receiver operating characteristic (AUROC) as Kappa and AUROC measure model accuracy, while compensating for simple chance [79]. The primary metrics we used from the confusion matrix included sensitivity (SE), specificity (SP), and positive predictive value (PPV) as these parameters give us the true positive rate, true negative rate, and the proportion of true positives amongst all positive calls [80–82]. Chemical-concentration combinations were defined as abnormal if the autoencoders identified more individual as abnormal in the exposed than their respective controls and at least 25% of the individuals were abnormal. Permutation feature importance was used to evaluate which features are the most important for model performance. In brief, one feature (variable) is shuffled randomly and all features are fed into the model the resulting Kappa and AUROC values are calculated. This is repeated 1000 times per feature and average Kappa and AUROC are calculated across each shuffle [83]. To determine why one feature might be more important than another a coefficient of variation was calculated for each of the features in the control and exposed groups (variation() in the SciPy package).

## Experimental confirmation of autoencoder findings

Following model development two chemicals were identified for follow-up laboratory testing. We generated new data using 2-Methylphenanthrene, and Nonafluoropentanamide. 2-Methylphenanthrene was chosen as the autoencoder identified it was different from controls at a much lower concentration than a K-S test of distance moved and angular velocity while Nonafluoropentanamide was selected as it was not identified using either a K-S test of distance moved and angular velocity. Similarity between the results was determined by comparing fourth order polynomial curve fits with and a significance threshold of $p < 0.05$.

## Supporting information

**S1 Table. Study chemicals and their common use.**
(XLSX)

**S2 Table. Statistical results for behavioral response analysis.**
(XLSX)

**S1 Fig. Loss function results during training.** Changes of loss functions during the training of (A) light-hypoactive controls, (B) light-normal controls, (C) light-hyperactive controls, (D) dark-hypoactive controls, (E) dark-normal controls, (F) dark-hyperactive controls. Blue line–training data (controls-only), orange line–test data (abnormal-only).
(TIF)

**S2 Fig. Interexperimental behavioral response to light/dark cycling in control larval zebrafish.** Zebrafish larvae were statically exposed to a chemical from six hpf until five dpf. At five dpf, behavior was measured under environmental conditions of continuous light for three minutes (0–180) followed by three minutes of dark (180–360). This plot shows representative control behavior data (n = 7 per line) classified as hyperactive (blue line), normal (green line) or hypoactive (purple line). The insert shows an example of larval behavioral tracks produced by Ethovision XT software. Figure depicts means ± SEM.
(TIF)

## Acknowledgments

We would like to thank the staff at Sinnhuber Aquatic Research Laboratory, and John Lam for his contribution to reprocessing videos.

## Author Contributions

**Conceptualization:** Adrian J. Green, David M. Reif.

**Data curation:** Lisa Truong.

**Formal analysis:** Adrian J. Green, Preethi Thunga, Melody Hancock.

**Funding acquisition:** Robyn L. Tanguay, David M. Reif.

**Investigation:** Lisa Truong, Connor Leong.

**Methodology:** Adrian J. Green.

**Project administration:** David M. Reif.

**Resources:** Robyn L. Tanguay, David M. Reif.

**Supervision:** Robyn L. Tanguay, David M. Reif.

**Validation:** Adrian J. Green, Lisa Truong, Connor Leong.

**Writing – original draft:** Adrian J. Green.

**Writing – review & editing:** Adrian J. Green, Lisa Truong, Preethi Thunga, Connor Leong, Melody Hancock, Robyn L. Tanguay, David M. Reif.

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
