## [Decision Letter · Decision Letter 0]

15 Dec 2023

Dear Dr. Reif,

Thank you very much for submitting your manuscript "Deep autoencoder-based behavioral pattern recognition outperforms standard statistical methods in high-dimensional zebrafish studies" for consideration at PLOS Computational Biology.

As with all papers reviewed by the journal, your manuscript was reviewed by members of the editorial board and by several independent reviewers. In light of the reviews (below this email), we would like to invite the resubmission of a significantly-revised version that takes into account the reviewers' comments.

We cannot make any decision about publication until we have seen the revised manuscript and your response to the reviewers' comments. Your revised manuscript is also likely to be sent to reviewers for further evaluation.

Sincerely,

Samuel V. Scarpino

Academic Editor

PLOS Computational Biology

Natalia Komarova

Section Editor

PLOS Computational Biology

Reviewer's Responses to Questions

**Comments to the Authors:**

Reviewer #1: Overall the authors did a could job at highlighting why one would want to opt to use deep autoencoders when analyzing behavioral data. A few sections were hard to follow. For example, the authors refer "traditional statistical methods" throughout the manuscript and vaguely comment on some of the caveats of the such methods for analyzing toxicological behavioral data. But the latter is done only at a superficial level. The authors don't put forward enough the intricacies of the toxicological behavioral data. What is particular about those data? Moreover, what are the specific caveats of the traditional statistical methods? Is one of the main findings of this study that autoencoders helped to discriminate between normal and abnormal behaviors for even very low concentration of toxins?

Below I provide further details of the aforementioned sections that could be rephrase to both improve flow and clarity of the manuscript -- especially for readers who might not be familiar to such data or methods.

L19: I would suggest to change to "model organism" rather than tool.

L75: How is this finding specifically crucial for public health? Be more specific

L81-82: "...making it applicable to multiple laboratories" Do the authors mean that due the high flexibility of autoencoders, other people working on similar type of highly variable data would be able to apply similar network architecture to their data?

L138: what does "dpf" stands for?

L138-142: It would be great if the authors could be more specific by elaborating how anomaly detection could potentially solve that issue mentioned in the example rather than vaguely mentioning that this issue could be tackled by machine learning. Perhaps the authors can specify the exact issues, from an analytical point of view, how does the traditional methods not account for them? How does anomaly detection provide an alternative path to solve this issue?

L158: "traditional statistical methodologies" - this is very vague, which methods are the authors specifically referring to?

L231: Do the authors refer to K-S test as the traditional statistical analysis?

L354 -- L363: In general I had a hard time understanding how the K-S test were performed? Did the authors compare the empirical cumulative distributions of individuals from different groups (hypo-, normal, hyperactive) to their controls to classify whether they were normal vs abnormal? What is the threshold of "normal" v/s "abnormal" behaviors?

Reviewer #2: The authors present a timely study concept of the implementation of autoencoder-based pattern recognition that is described as an improvement upon standard statistical methods of classification of zebrafish exposed to various neurotoxic chemical treatments. Despite a useful discussion of the need for more sophisticated analysis methods to be applied to behavioral classification as these techniques are increasingly implemented in various fields, the authors have not established a correct standard in the field for the comparison base that they use for their main analysis design.

A major drawback of the design of the study is that the authors frame all improvements with their autoencoder method against basic statistical methods that are applied to classification based upon a metric of distance traveled, velocity, x-position, and y-position. Since each of these 4 measures are discussed, it is unclear to the reader whether all 4 of these measures were used for the statistical model or if only distance traveled is used. Please clarify this in the revision.

This measure of distance traveled is not the typical standard in the field for classification of subjects that are treated with toxins. Rather, distance traveled is often utilized as a locomotor control, which often is not capable of detecting variation between subjects treated with a toxin.

If locomotor impairment in the form of reduced distance traveled is found, this standardly indicates that subjects are not able to demonstrate more complex effects of toxin exposure. Rather reduced distance traveled would show that subjects display gross locomotor impairment, which could be attributed to multiple effects. This gross locomotor impairment could be due to inhibition of motor control or behavioral impacts and cannot be attributed to either effect as they are confounded with one another when dose of toxin is high enough to induce locomotor impairment. For the reasons discussed above, other metrics of behavioral consequences of toxin exposure including angular velocity and turn angle are utilized by many of toxicological studies utilizing the zebrafish model as they are sensitive to toxin effects, which may otherwise not be measured using metrics of distance traveled and velocity alone.

This standard of comparison of the autoencoder method against the distance traveled / velocity metric is flawed for these reasons. As such, it would be a great improvement upon the basic points of this study if a more standard metric such as angular velocity, or turn angle is used as the comparison base for classification using standard statistical methods. Otherwise, it is already expected that the distance traveled metric should not vary across subjects if the subjects are receiving a dose of toxin that is known to cause a behavioral impact without gross locomotor impairment (since distance traveled is a locomotor control and is not suggested as a metric for differentiating subjects using standard methods).

The basic point of the deep autoencoder improving upon standard methods would be greatly strengthened if the authors perform an analysis based on metrics that are more established for assessing subtle toxic effects (that are shown even when distance traveled does not change, such as changed angular velocity or changed turn angle, compared to controls.) Please address some form of an analysis of metrics other than distance traveled and velocity as the basis of comparison against the autoencoder-based strategy this in revision of this manuscript.

Is there any improvement of the autoencoder strategy compared to metrics such as angular velocity and turn angle? If there is no additional improvement from the autoencoder strategy when using these more typical measurements of subtle toxic effects, then the authors should declare this in the manuscript for full transparency and for generalizability of their findings to the field as a whole, in which these measures are commonly utilized.

I have discussed this point in detail since it is essential to address to improve the theoretical basis of the study design. Aside from this point, there are some additional points below. It is uncertain how the doses of toxins were chosen in this study. Were they based on the literature from other researchers or arrived at due to preliminary / unpublished studies within the authors’ lab or previously published work within the authors' lab. In either case, it should be specifically cited how these doses were arrived upon and whether the authors assessed standard metrics for establishing dose such as LD50 for each toxin. Please address this in revision of this manuscript.

It is unclear to the reader how the authors established the subjects as hyperactive, normal, or hypoactive. Was this based on categorical splitting of the data into equally numbered groups as is done with a traditional median split? Or were these classes based on specific cut-off points of the subjects based on other studies in the field which have standard cut-offs for these metrics? Please address this in revision of this manuscript.

The discussion section would benefit from some explanation of the purported reasons why certain classes of chemicals are better differentiated by the autoencoder-based method vs. standard statistical methods. Does this vary according to chemical class / structure?

It is unclear what type of analysis was performed in Figure 4 to determine that the trained model and the test dataset’s model produce similar results or if this is simply a qualitative observation that the training and test data models seems similar to the authors. It would be valid to claim similar results on the basis of a non-significant statistical test, rather than just a visual inspection.

Some editorial changes: On the referenced GitHub page from lines 384-385, the Autoencoder Model Design and the ten Jupyter Notebook file links appear to be broken; In line 29 should be “abnormal behavioral effects” instead of “abnormal behavioral”

Based on the need to sufficiently address these points described above, I would recommend resubmitting the manuscript with major revision.

Reviewer #3: This manuscript submitted by Green and colleagues reports a method based on machine learning to improve identification of abnormal behaviour of zebrafish larvae. Considering that behaviour of zebrafish is a frequently used methods to evaluate (neuro)toxicity of chemicals and the large amount of chemicals not tested yet, any way to improve sensitivity or accuracy is of utmost importance.

The Ms is well organised and easy to read (this reviewer not being able to evaluate the computational part focused on the behavioural and fish parts) and globally acceptable for publication provided below comments are addressed. One point would deserve more in-depth discussion because some results are puzzling. Indeed, in some cases, a higher sensitivity can explain differences between statistical and autoencoder methods (Table 2), but in other ones (Multi-Walled Carbon Nanotube; 1-Methylphenanthrene, Dibenzo[e-l]pyrene, Aminohexafluoropropan-2-ol) differences are in the middle of the concentrations range. How the Authors can ensure they are not false positive?

Given that the procedure appears to require high-level computational expertise, how can this help as many labs as possible benefit from this new approach?

Specific comments

L116 The Authors write "Zebrafish larvae show mature swimming patterns … at four to five days post-fertilization (dpf), which can be assessed using various locomotor behavioral assays…"

The sense of "mature swimming pattern" is not clear; it does not correspond to a specific situation. In addition 1) behaviour is very variable between 4 and 6 dpf and 2) response to chemicals varies significantly with often no response at 4 dpf while defect can be detected at 5 dpf. The sens of mature should be precisely defined (or the sentence rewritten) and 4 to 5 dpf should be distinguished in terms of maturity of responses.

L256 "more recently" is not correct since most studies are older

L297 reference is made to Fig. 3 while ti should be Fig. 2.

L318 type of 96 well plates should be indicated since round well plates have proven to produce high occurrence of swim bladder inflation defects. According to figures round well plates were used, so percentage of malformed larvae (see next comment) should be provided. Medium renewal or its absence should be clearly indicated.

L324 morphological defects scored at 24 and 120 hpf should be indicated

L331-332 the recording protocol is not correctly described, 3 cycles of 3+3 phases are equivalent to 18 minutes and not 24. Perhaps because acclimation precedes the first cycle. This should be corrected, and for acclimation if any, was it in the light or in the dark?

L335-336 so what is the purpose of this third cycle?

L544 ref format not correct

Even though Tanguay's laboratory has produced a large amount of data dealing with the behavior of zebrafish, a fairer representation of the literature should be presented!

**Have the authors made all data and (if applicable) computational code underlying the findings in their manuscript fully available?**

Reviewer #1: Yes

Reviewer #2: **No: **Two links on Github appear to be broken upon attempt to access: On the referenced GitHub page from lines 384-385, the Autoencoder Model Design and the ten Jupyter Notebook file links appear to be broken

It is stated that the data is available upon request instead of being uploaded to a public repository.

Reviewer #3: Yes

PLOS authors have the option to publish the peer review history of their article (what does this mean?). If published, this will include your full peer review and any attached files.

Reviewer #1: No

Reviewer #2: No

Reviewer #3: No
---

## [Decision Letter · Decision Letter 1]

6 May 2024

Dear Dr. Reif,

Thank you very much for submitting your manuscript "Deep autoencoder-based behavioral pattern recognition outperforms standard statistical methods in high-dimensional zebrafish studies" for consideration at PLOS Computational Biology. As with all papers reviewed by the journal, your manuscript was reviewed by members of the editorial board and by several independent reviewers. The reviewers appreciated the attention to an important topic. Based on the reviews, we are likely to accept this manuscript for publication, providing that you modify the manuscript according to the review recommendations.

Sincerely,

Samuel V. Scarpino

Academic Editor

PLOS Computational Biology

Natalia Komarova

Section Editor

PLOS Computational Biology

Reviewer's Responses to Questions

**Comments to the Authors:**

Reviewer #2: Thank you for including thorough responses to the majority of the issues that I addressed in my initial review. The few items remaining to address are included below:

Supplementary figures or table information should be included to display the directionality of the effects described in Supplementary Table 2 and Supplementary Table 3. As is, it is unknown whether the chemicals cause an increase or decrease in each of the distance traveled and angular velocity measures for each chemical. Please include figures showing this or information in a supplementary table showing the mean and standard deviation (or standard error) and number of samples for each chemical and control levels for both distance traveled and angular velocity.

Please address the data availability issue described in the data availability section. I have included this here for reference: The full set of data and code are not provided as part of the manuscript or its supporting information or deposited to a public repository. Although part of the data are provided the manuscript states that you need to contact the authors to access the full dataset and code. This does not yet meet the PLOS Data policy description.

Reviewer #3: Thanks for having addressed most of the comments raised. Could you elaborate a bit about the way other laboratories will be able to adopt this approach on the one hand and on the other hand the flexibility of the behavioral protocol (since there is no standard protocol). Second point please provide information about lighting during acclimation.

**Have the authors made all data and (if applicable) computational code underlying the findings in their manuscript fully available?**

Reviewer #2: **No: **The full set of data and code are not provided as part of the manuscript or its supporting information or deposited to a public repository. Although part of the data are provided the manuscript states that you need to contact the authors to access the full dataset and code. This does not yet meet the PLOS Data policy description.

Reviewer #3: None

PLOS authors have the option to publish the peer review history of their article (what does this mean?). If published, this will include your full peer review and any attached files.

Reviewer #2: No

Reviewer #3: No

Figure Files:

Data Requirements:

Reproducibility:

References:

---

## [Editor Report · Decision Letter 2]

15 Aug 2024

Dear Dr. Reif,

We are pleased to inform you that your manuscript 'Deep autoencoder-based behavioral pattern recognition outperforms standard statistical methods in high-dimensional zebrafish studies' has been provisionally accepted for publication in PLOS Computational Biology.

Best regards,

Samuel V. Scarpino

Academic Editor

PLOS Computational Biology

Natalia Komarova

Section Editor

PLOS Computational Biology

---

## [Editor Report · Acceptance letter]

31 Aug 2024

PCOMPBIOL-D-23-00581R2 

Deep autoencoder-based behavioral pattern recognition outperforms standard statistical methods in high-dimensional zebrafish studies

Dear Dr Reif,

I am pleased to inform you that your manuscript has been formally accepted for publication in PLOS Computational Biology. Your manuscript is now with our production department and you will be notified of the publication date in due course.

With kind regards,

Zsofia Freund
